# Prediction of Depression for Undergraduate Students Based on Imbalanced Data by Using Data Mining Techniques

**Warawut Narkbunnum and Kittipol Wisaeng ***

Mahasarakham Business School, Mahasarakham University, Mahasarakham 44150, Thailand
* Correspondence: kittipol.w@acc.msu.ac.th

**Abstract:** Depression is becoming one of the most prevalent mental disorders. This study looked at five different classification techniques to predict the risk of students' depression based on their socio-demographics, internet addiction, alcohol use disorder, and stress levels to see if they were at risk for depression. We propose a combined sampling technique to improve the performance of the imbalanced classification of university student depression data. In addition, three different feature selection methods, Correlation, Gain ratio, and Relief feature selection algorithms, were used for extracting the most relevant features from the dataset. In our experimental results, we discovered that combining the bootstrapping technique with the Relief selection technique under sampling methods enabled the generation of a relatively well-balanced dataset on depression without significant loss of information. The results show that the overall accuracy in the risk of depression prediction data was 93.16%, outperforming the individual sampling technique. In addition, other evaluation metrics, including precision, recall, and area under the curve (AUC), were calculated for various models to determine the most effective model for predicting risk of depression.

**Keywords:** data mining techniques; prediction; depression; imbalanced data; feature selection

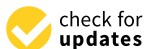



## 1. Introduction

Depression is a common illness; more than 300 million people worldwide suffer from it. Depression can start at any time and affect anyone. It may occur once or on a recurring basis. The presence of persistent sadness and a loss of interest in normally enjoyable activities, along with an inability to perform daily tasks for at least two weeks, characterize depression. It is different from normal mood swings or short-term sadness caused by the problems of everyday life [1]. Furthermore, Thailand is experiencing an increase in the incidence of adolescent depression. Depression is 14.9% prevalent among Thai teenagers [2]. Depression has a direct detrimental impact on the quality of life of teenagers, including unpleasant emotions such as sadness, remorse, and low self-esteem. Data revealed that in 2017, the suicide rate among 20- to 24-year-olds was 4.94 per 100,000 people. In 2018, it increased to 5.33 per thousand people. According to yearly data, youth groups increasingly called the Mental Health Hotline 1323 to discuss mental health concerns. During 2018, 70,534 telephone consultations were provided, including 10,298 consultations with children aged 11 to 19 (14.6%) and 14,173 consultations with youth aged 20 to 25 (20.1%). Stress and anxiety are among the top five most prevalent issues among children and adolescents. Regarding psychiatric problems during the first six months of the fiscal year 2019, the service received 40,635 calls, including 13,635 calls from adolescents and young adults aged 11 to 25, according to Mahasarakham University students who visit mental health centers for treatment. During the academic years 2013 to 2017, students with major depressive disorders were diagnosed with 233 psychiatric disorders, or 36.5% [3].

The Depression Screening Question is a basic and time-consuming questionnaire that can be used for depression assessment and screening. If a patient has a depressed entrance score, an 8-question suicide assessment (8Q) is necessary to assess and continue managing

the patient appropriately. Depression screening is, therefore, crucial and straightforward. It employs brief questions and 8Q and 9Q [4–6] tests to identify patients with depressive disorders and appropriately introduce them to the program. Although depression screening tools vary in their psychometric properties, the PHQ-9 is the most extensively psychometrically-tested tool [7]. There is a substantial number of standard questionnaires available in the Thai language, for example the Patient Health Questionnaire (2Q, 8Q, 9Q, and PHQ-A) [8–10] and the Beck Depression Inventory-Thai [11].

Knowledge of data mining is currently utilized in various fields of research, including medical research, as it examines techniques used to classify data in other domains to conduct research and utilize them to develop models for screening students at risk of depression [12–14]. Data mining in medicine is an emerging discipline of tremendous relevance for providing a prediction and a deeper understanding of illness classification, particularly in mental health domains concerning the most prevalent diseases of Alzheimer's, schizophrenia, and depression. In terms of percentages, the main techniques applied to depression are support vector machines [15–17], followed by Naïve Bayes, Random Forest [18], logistic regression [12,19,20], and the Decision Tree [21]. It has been shown that data mining techniques are increasingly being used to predict depression in students [22–25]. Using data mining techniques and specific clinical and demographic factors that predict, we can identify depression risk in a timely and cost-effective manner. Some of these studies combine socio-demographic factors with physiological factors obtained from medical tests, which are also compatible with biomarker diagnosis [26–28].

In conclusion, data-mining techniques are accessible and effective diagnostic aids that we can include in the diagnostic workflow. Implementing them in public health care systems or by reaching out to the public, for example, by disseminating triggering advertisements offering to scan people's social media and alert them if they need to seek help, can be excellent ways to alert people about a possible risk for depression [29].

The goal of this study was to determine whether a person is at risk of developing depression, to identify the key factors that cause depression, and to determine the best machine learning approach to identify at-risk people. This study also intended to minimize the required time for screening depression.

The major contributions to these goals are as follows:

1.  Identifying the most important socio-demographic, internet addiction, alcohol use disorder, and stress factors of students that contribute to depression formation;
2.  Creation of a dataset containing the students' socio-demographics, internet addiction, alcohol use disorder, and stress to predict depression;
3.  Exploring different data mining and feature selection algorithms to efficiently screen for the existence of depression;
4.  Due to the simplicity of the required socio-demographic, internet addiction, alcohol usage disorder, and stress information of students used in this study, a student suspected to be at-risk for depression would feel less hassle giving the required information of this study to detect depression rather than answering the questions of different authentic depression screening scales.

## 2. Related Work

### 2.1. Depression

Depression is a mental health issue that affects people worldwide, ranking third after cardiac and respiratory diseases as a major cause of disability, particularly among university students [30]. In Thailand, one of Chiang Rai's higher education schools was surveyed regarding the prevalence of depression among university students in years 1–4, ages 18 to 22. Using the Center for Epidemiological Studies-Depression Scale (CES-D), researchers discovered a prevalence of 31.9%. The greatest prevalence was in the Nursing Faculty [31]. Students in the first year of study at the Suranaree University of Technology responded to online questionnaires. The study revealed that 17.0% of students were depressed and 51.10% were stressed [32]. Among preclinical medical students, the

University of Ubon Ratchathani found a 32.87% prevalence of depression considered mild, 27.97% were somewhat depressed, 4.20% were seriously depressed, and 0.70% enrolled in nursing school [33].

We should identify risk factors associated with depression among university students early in university to provide them with additional mental health support and prevent the worsening of risk factors.

### 2.2. Demographic Information

Many studies have established that the prevalence rate of depression varies from 2.8% to 10.3% based on socio-demographic factors [26]. The variables that have most often been studied in relation to mental health literacy are gender, college major, grade year, GPA, homeland [34], socio-economic factors [35], family status [36], health condition, and history of depression in the family [37].

### 2.3. Internet Addiction

In these modern times of digitization, the internet has become an integral part of everyday life, especially the lives of adolescents. Internet addiction is a severe problem with a profound impact on mental health [38–40]. The beginning of the twenty-first century has witnessed phenomenal growth in internet usage, particularly in developing nations such as Thailand [41]. As examined by this review, the impact of social media use on the incidence of depression, anxiety, and psychological distress among adolescents is likely to be multifactorial. The key findings of the included studies were categorized into four types of social media exposure: time spent, activity, investment, and addiction [40].

Internet addiction is an emerging concern, but there is a lack of comprehensive understanding of the causes of problematic behavior and of a gold-standard method for assessing symptoms. This study aimed to perform a psychometric analysis utilizing the most widely used screening instrument, the Young Internet Addiction Test (IAT) [42,43].

### 2.4. Alcohol Usage Disorder

Depression is characterized by fluctuating emotions. According to a literature review on psychosocial factors of depression associated with the risk of suicide in adolescents, several studies have been conducted in Thailand. Depressive disorders were found to be up to 14.3% of the factors associated with adolescent suicide, and reflections on the association between depression and suicide risk among secondary school-aged adolescents were conducted [44]. Depression was found to positively correlate with a statistically significant risk of suicide. The same holds for international research studies. In addition to concluding that it is important and necessary to study the correlation between depression and suicide risk, the review found that alcohol consumption was significantly statistically correlated with suicide risk [45,46]. Alcohol consumption is a major risk factor for university students [47] and depression is also positively associated with alcohol-related behaviors. Alcohol use disorder (AUD) and alcohol abuse are prevalent among adolescents and young adults and are linked to serious personal and social problems.

This study employed the Alcohol Use Disorders Identification Test (AUDIT), which is a simple and effective screening instrument for unhealthy alcohol use, which is defined as risky or harmful intake or any alcohol use disorder [48,49].

### 2.5. Stress

Daily stress is unavoidable. It is possible for anyone to experience it. It serves to improve awareness, promote learning, and increase productivity at work. Due to the need to adjust to rapid changes in physical, emotional, and social contexts, adolescence relates to an increased risk of mental health issues. Stress, anxiety, sadness, and suicide are all prevalent psychological issues in adolescents. Mental health difficulties affect adolescents in various ways, including the consequences of stress on self-efficacy. Students that are under a great deal of stress have low self-efficacy. Excessive stress can have a negative

impact on both physical and mental health, self-esteem, and the success of learning and self-development. Teenagers are affected by mental health issues in a variety of ways, including the effects of stress on self-efficacy. One of the leading causes of suicide is depression. Financial restrictions, a lack of social connection, and concerns about transitioning to university life are connected with higher stress and anxiety among students [50].

This study, using a self-administered online questionnaire, attempted to quantify students' stress levels and identify factors that influence depression. The stress measurement instrument was adapted from the Department of Mental Health, Ministry of Public Health's stress evaluation form using the self-reported stress test (Suan Prung Stress Test, SPST-20) [51,52].

## 3. Materials and Methods

### 3.1. Instruments

The researchers followed the following procedures and methodologies:

1. Created a questionnaire based on the quantity of samples used for research and made sure the document was correct, complete, and had all the information it needed;
2. Asked for a formal letter from the university;
3. Collected information from each faculty; a process called "informed consent" was used to get the subjects' consent and make sure they understand what is going on;
4. Held the processing period from June 2020 through March 2021;
5. Examined the responses to the questionnaire for completeness;
6. Collected information from the received questionnaires for further data analysis and interpretation.

The questionnaire consisted of five types, categorized as Type A–E, containing a total of 69 questions, and all questionnaires were non-identifiable to the participants. All questionnaires were filled out by the samplers and required between 20 and 45 min to complete. Type A–D consisted of 59 questions to be used as a predictive variable for depression risk, and Type E consisted of 9 questions to be used as a target variable. Each section's questionnaire was as follows:

### 3.1.1. A Demographic Information Questionnaire (Type A)

In addition to relevant reviews, a personal information questionnaire that included questions about gender, college major, grade year, GPA, homeland, income adequacy, family status, health condition, and history of depression in the family.

### 3.1.2. Assessment of Internet Addiction (Type B)

The social network usage questionnaire adapted from the Young Internet Addiction Test was separated into three categories: frequency of use, length of use, and social network addiction. The test consisted of 12 questions using a Likert scale, which is used to determine the degree of agreement. The total IAT score was the sum of the examinee's assessments for the 20-item responses. Each item was evaluated from 0 to 5 on a 5-point scale (from 1—not at all, to 5—always). The maximum possible score was 100. The severity of the problem increases as the score increases. Scores between 0 and 30 indicate Internet addiction, 50 to 79 show a moderate level, and 80 to 100 indicate a severe reliance on the Internet.

### 3.1.3. Alcohol Use Identification Test (AUDIT) (Type C)

The World Health Organization created this evaluation as a time- and space-constrained screening tool for excessive binge drinking. It can also assist in identifying problems with binge drinking that result in illnesses that send people to the doctor. The evaluation is a self-evaluation. It takes some time, but the evaluation is simple.

The AUDIT consists of 10 questions, each with a score of 0, 1, 2, 3, or 4, except for questions 9 and 10, which carry scores of 0, 2, and 4. The range of scores is from 0 to 40, with 0 indicating/showing an abstainer who has never experienced alcohol-related difficulties. The World Health Organization (WHO, Geneva, Switzerland) recommends a score of 1, and

7 indicates low-risk usage. Eight to fourteen points indicate potentially hazardous alcohol usage, while a score of fifteen or higher suggests the likelihood of alcohol dependence (moderate-severe alcohol use disorder).

### 3.1.4. Stress Test (Type D)

Thailand's Department of Mental Health (2018) developed a stress test to evaluate what had happened over the previous six months. The SPST explores feelings over the previous 6 months, asking about occurrences and feelings towards such occurrences. The assessment criteria were as follows: 1 point of the stress score represents no stress, 2 points of the stress score represents low stress, 3 points of the stress score represents moderate stress, 4 points of the stress score represents high stress, and 5 points of the stress score represents the highest level of stress. Total scores were summed up and compared with the criteria for assessment of stress level as follows: 0–24 points indicate low stress, 25–42 points indicate moderate stress, 43–62 points indicate high stress, and more than 63 points indicates severe stress.

### 3.1.5. Patient Health Questionnaire for Adolescents (PHA-A)

A screening model called the Patient Health Questionnaire for Adolescents (PHQ-A) was created to evaluate depression in young people for the use of Thai teenagers. It is a brief examination, but adolescent depression assessments are accurate and congruent with research using earlier, original English-language instruments, including PHQ-9 and PHQ-A. A self-answer questionnaire with 9 items to gauge the severity of depressive symptoms, it is simple to use, reliable, and accurate when measuring depression in young people [53].

### 3.2. Data Set

The collection of data is a crucial component of model development. There are a limited number of data collection methods that are entirely dependent on the type of research being conducted. Observation, interview, document scanning, measurement, questionnaire, or a combination of these methods may be used to collect data [15]. In our work, data was collected from a questionnaire prepared by the authors.

According to the division of registration at Mahasarakham University, there were 37,579 undergraduate students registered for the second semester of the 2019 academic year. The sample size was calculated using Krejcie and Morgan's pre-made table, which established the population's proportion of desirable qualities, 0.5% acceptable tolerance levels, and 95% confidence intervals for 380 individuals.

The survey was conducted in the period between June 2020 and March 2021. The data set consisted of the responses of 460 participants. The inclusion criteria for this study were: either male or female, outpatients, aged more than 18 years, diagnosed without mood disorders, and volunteers who did not know they had depression. Participants were excluded if they were under the age of 18 years, illiterate, enrolled in other trials, or were in situations that did not allow obtaining written informed consent. Participants were not paid. The contents of the monitoring interviews were reviewed to identify patients who had attended at least 2 appointments. These criteria yielded the 380 participants

Among the 380 participants in the accumulated dataset, 296 participants were found to be at risk for depression. The prevalence of depression among the participants of the survey was 77.9%. Table 1 shows the distribution of depressed and non-depressed participants in the dataset.

**Table 1.** Distribution of depressed and non-depressed dataset participants.

| Class | Number of Participants | Percentage (%) |
| :---: | :---: | :---: |
| Depressed | 296 | 77.9 |
| Non-depressed | 84 | 22.1 |

Data in Table 1 are represented with imbalanced data, which is common when predicting with big data. Raw data are directly analyzed without using supplementary techniques such as a sample algorithm for data sets with imbalanced class ratios. This can decrease the performance of machine learning by causing prediction errors. Imbalanced data are data in which certain groups are overrepresented. One of the groups has a large amount of data, separated into majority and minority classes. Imbalanced information will result in the incorrect or less accurate categorization of the data of the minority group. However, it is possible to categorize the data of the majority group more precisely. General information can contain unbalanced information.

In order to improve the data before analyzing them, the Synthetic Minority Over-sampling Technique (SMOTE) [54–57] and resampling with replacement (bootstrapping) [58–60] were used in this study. With SMOTE, numerous variations exist for an oversampling technique that generates synthetic data by approximating minority instances. It is a type of oversampling method that has been demonstrated to be effective and is frequently utilized in machine learning with imbalanced high-dimensional data, and is increasingly utilized in the medical field. As an alternate way of assessing mediation, bootstrapping is a non-parametric resampling procedure that does not assume the normality of the sampling distribution. Bootstrapping is a computationally demanding method that estimates the indirect effect of each resampled data set by repeatedly sampling from the data set. In addition, as part of the preprocessing, the interviewer's pauses and background noise were deleted.

This study prepared the data for sampling using the following methods: Table 2 shows the original data (D-ORI), the minority oversampling data (D-SM), and the resample with replacement data (D-RE).

**Table 2.** Research data set after data preparation.

| Data Input | Instances | Majority (Yes) | Minority (No) | Imbalance Ratio (maj/min) |
|------------|-----------|----------------|---------------|---------------------------|
| D-ORI | 380 | 296 | 84 | 3.52 |
| D-SM | 464 | 296 | 168 | 1.76 |
| D-RE | 380 | 296 | 84 | 3.52 |

From Table 2, it was determined that D-ORI had 380 instances, where majority or those at-risk were recorded for 296 cases and minority or those not at-risk were recorded for 84 cases, demonstrating that the imbalance between majority and minority equals 3.52. D-SM had 464 instances, with a majority of 296 and a minority of 168, for a ratio of 1.76, while D-RE had 380 instances, with a majority of 296 and a minority of 84, for a ratio of 3.52.

### 3.3. Target Variable

Informing the status of depression as screened by the PHQ-A questionnaire for measuring the target variable: if any of these variables' value is true, the target variable is defined as "risk of depression" = 1, otherwise "non-risk of depression" = 0.

### 3.4. Feature Selection Techniques

During the construction of a model, only essential components should be chosen. The model's performance may degrade if irrelevant features are selected. Selecting inappropriate characteristics may result in a deterioration of the model's performance. Feature selection makes it easier to get rid of features that are not needed or are redundant and do not change how well the model works.

The following three strategies of characteristics evaluation were employed in the current study: CorrelationAttributeEval, GainRatioAttributeEval and ReliefFAttributeEval. The feature search strategy made use of the ranker method, and the following search techniques were implemented using the Weka software as described below:

CorrelationAttributeEval: This option can evaluate the worth of an attribute by measuring the correlation (Pearson's) between it and the class;

GainRatioAttributeEval: This option can evaluate the worth of an attribute by measuring the gain ratio with respect to the class;

ReliefFAttributeEval: This option can evaluate the worth of an attribute by repeatedly sampling an instance and considering the value of the given attribute for the nearest instance of the same or different class that can operate on both discrete and continuous class data.

### 3.5. Data Mining Techniques for Depression Prediction

To predict depression, this study used 5 different data mining classifiers, namely: the Support Vector Machine (SVM), Naïve Bayes (NB), Logistic Regression (LR), Random Forest (RF), and Decision Tree (J48).

#### 3.5.1. Support Vector Machine

SVM is one of the methods in the problem of pattern classification. SVM finds the optimal hyperplane that categorizes training data input into two classes (good and bad). Figure 1 shows the basic support vector machine.

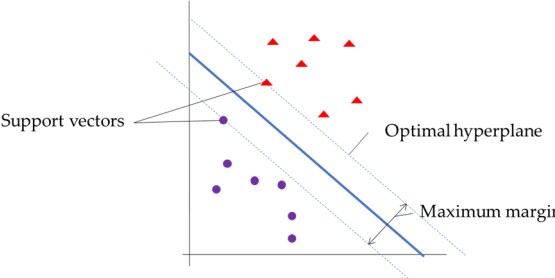

**Figure 1.** Base of support vector machines.

The SVM algorithm is based on the kernel method, and the selection of the kernel type has a strong effect on the classification results. The type of kernel used was the Radial Basis Function (RBF) kernel, as in Equation (1):

$$K(x, x') = exp\left(-\gamma \frac{\|x - x'\|^2}{\sigma^2}\right) \tag{1}$$

#### 3.5.2. Naïve Bayes

NB is a simple, supervised learning method for classification by calculating the probability to infer the solution. A conditional probability model is used as the data training model. It is appropriate for classifying a large sample, as calculated in Equation (2):

$$P(A|B) = \left(\frac{P(B|A), P(A)}{P(B)}\right) \tag{2}$$

#### 3.5.3. Random Forest

RF is an ensemble learning technique. Multiple sets of training data and unique features are selected at random. The models are then constructed using a series of decision trees, where the out-of-bag data are collected for the prediction data test. Finally, the model outcomes are put to a vote, and the one that receives the most votes is the solution, as represented in Figure 2.

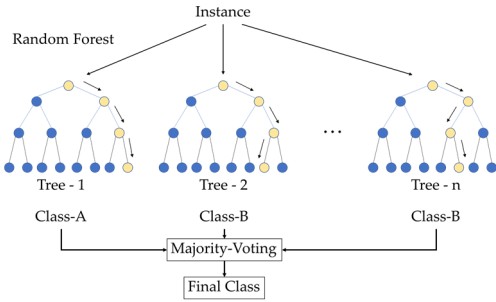

**Figure 2.** Typical Random Forest.

### 3.5.4. Logistic Regression

LR is applicable to data sets with dichotomous dependent and independent features. Due to the binary nature of the depression prediction data set, it cannot be modeled using linear regression. Logistic regression (LR) is necessary for such data. Two sets of experiments exist. One set produces a positive result, which is the attribute set by which depressive people are identified, while the other produces a negative result, which identifies those who are not depressed. The logistic function used in this model to predict the output of an experiment is illustrated in Equation (3).

$$f(r) = \left( \frac{1}{1 + e^r} \right) \tag{3}$$

### 3.5.5. Decision Tree

J48 is a classifier based on trees utilized for predictive modeling. It consists of an algorithm to construct decision tables and a visualization component to depict the model's graphical user interface. It creates a hierarchical structure of the input data set based on the relationship among the data. The algorithm selects the most important attribute set from the original data set, reducing the likelihood of overestimation. From this small and compact output decision table, a classification decision is made about how to set rules for making decisions among certain qualities, as shown in Figure 3.

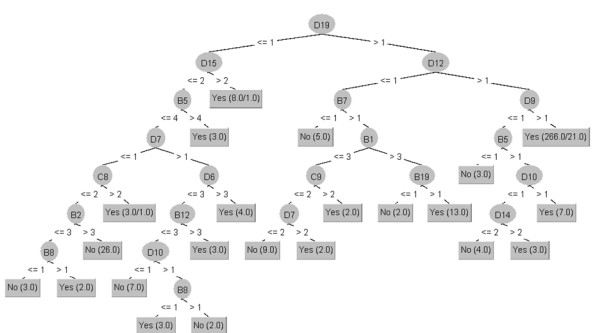

**Figure 3.** Classifier tree visualizer. Tree J48 from WEKA.

The configurations of the data mining classifier used in each experiment were determined in this study as shown in Table 3.

### 3.6. Training the Model

Following the selection of features, the models were built using the classification techniques. The 10-folds cross validation technique was used to validate the models' performances. In this technique, the entire dataset is divided into 10 subsets and then processed 10 times. Nine subsets are used as testing sets, and the remaining one is used as a training set. Finally, the results are shown by averaging each of the 10 iterations. The

subsets are divided using stratified sampling, meaning that each subset will have the same class ratio as the main dataset.

**Table 3.** Parameters of the data mining techniques.

| Technique | Parameter Setting |
|-----------|-------------------|
| SVM | Batch size = 100, cache size = 40, kernel type = RBF, gamma = 0 |
| NB | Batch size = 100, numDecimalPlaces = 2 |
| LR | Batch size = 100, numDecimalPlaces = 4 |
| RF | Batch size = 100, numDecimalPlaces = 2, numExecutionSlots = 1, seed = 1 |
| J48 | Batch size = 100, confidence factor = 0.25, mumDecimalPlaces = 2, seed = 1 |

*3.7. Performance Evaluation*

Different performance metrics such as accuracy, precision, recall, and area under the curve (AUC) having been determined for all the models constructed in the previous subsection, the efficacies of these models were evaluated based on these performance metrics. Finally, the best model for predicting depression was chosen, according to the outcomes of these performance metrics.

The proposed model used WEKA and a confusion matrix to estimate classification algorithms: SVM, NB, RF, LR, and J48. For the model, there were three most common evaluation indexes: accuracy, precision, and recall. The calculation of these evaluation indexes is inseparable from the existence of a confusion matrix, which is often used in classification algorithms. Accuracy reflects the proportion of correct classification in the classification results of each category; that is, the accuracy of each category is judged by the model. The formula for this is provided in Equation (4). The ratio of accurately identified true positives to total positive samples is known as precision. The formula is provided in Equation (5). The recall rate reflects the sensitivity of the classification model to each category dataset. The recall percentages of correctly identified positive samples, total positive samples, and total false-negative samples are illustrated in Equation (6). The Area Under the Curve is a measure of a classifier's ability to distinguish between classes and is used as a summary of the ROC curve. The higher the AUC, the better the model's performance at distinguishing between the positive and negative classes, as shown in Table 4.

$$Accuracy = \frac{total\,correct\,prediction}{total\,prediction}, \tag{4}$$

$$Precision = \frac{TP}{TP + FP}, \tag{5}$$

$$Recall = \frac{TP}{TP + FN}, \tag{6}$$

**Table 4.** The expert scale for AUC values shows the quality of the test.

| AUC Value | Test Quality |
|-----------|--------------|
| 0.90–1.00 | Excellent |
| 0.80–0.90 | Very good |
| 0.70–0.80 | Good |
| 0.60–0.70 | Satisfactory |
| 0.50–0.60 | Unsatisfactory |

Our model employed classification techniques such as SVM, NB, RF, LR, and J48 for prediction. Figure 4 presents the proposed model structure.

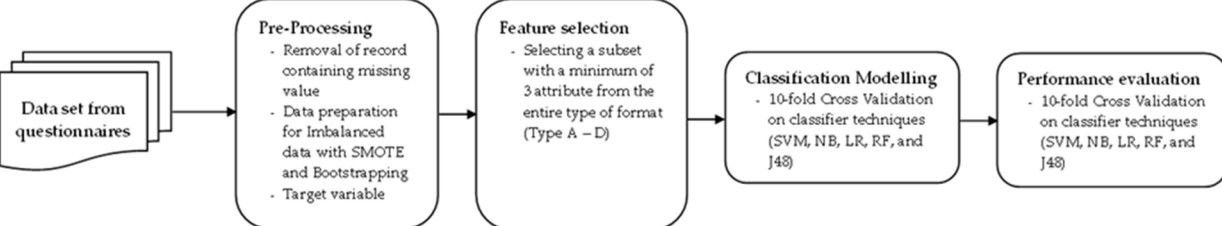

**Figure 4.** The flow diagram of the proposed model.

**4. Results**

*4.1. Data Analysis*

Most of the respondents were female (88.95%, *n* = 338), in their 3rd year of study (51.05%, *n* = 194), with a GPA greater than 3.01 (32.89%, *n* = 125), from other provinces (84.47%, *n* = 321), with an adequate income (62.63%, *n* = 238), whose family was together (73.42%, *n* = 279), with no health condition (87.11%, *n* = 331), and no history of depression in the family (90.53%, *n* = 344). According to research on depression among college students in Thailand, most of the people who answer the surveys are female [32,61–63]

*4.2. Internet Addiction Level, Alcohol Usage Identification Level, and Stress Level*

The following can be described as the results of the questionnaire's Parts B, C, and D: the degree of internet addiction was moderate for the majority at 75%, the level of alcohol use was predominantly low-risk at 66.3%, and the level of stress was predominantly high at 38.42% and severe at 33.68%

*4.3. Evaluation*

Tables 5–7 show the confusion matrices of the predicted results for each of the classifiers using a different feature selection technique. The measured accuracy, precision, recall, and area under the curve (AUC) of the classifiers for the constructed models are shown.

**Table 5.** Performance of the classifiers using the correlation technique.

| Balancing Technique | Classifier Technique | Accuracy (%) | Precision (%) | Recall (%) | AUC |
|---|---|---|---|---|---|
| Without Balancing | SVM | 82.37 | 0.846 | 0.946 | 0.669 |
| | NB | 76.84 | 0.913 | 0.777 | 0.841 |
| | LR | 83.16 | 0.865 | 0.929 | 0.838 |
| | RF | 79.21 | 0.847 | 0.895 | 0.782 |
| | J48 | 78.95 | 0.853 | 0.882 | 0.709 |
| SMOTE | SVM | 79.74 | 0.824 | 0.868 | 0.770 |
| | NB | 75.65 | 0.873 | 0.723 | 0.859 |
| | LR | 78.02 | 0.830 | 0.824 | 0.874 |
| | RF | 82.97 | 0.851 | 0.889 | 0.889 |
| | J48 | 78.02 | 0.834 | 0.818 | 0.784 |
| Bootstrapping | SVM | 83.16 | 0.852 | 0.949 | 0.683 |
| | NB | 77.63 | 0.914 | 0.787 | 0.861 |
| | LR | 81.84 | 0.860 | 0.916 | 0.863 |
| | RF | 90.53 | 0.928 | 0.953 | 0.946 |
| | J48 | 86.32 | 0.901 | 0.926 | 0.830 |

**Table 6.** Performance of the classifiers using the Gain Ratio technique.

| Balancing Technique | Classifier Technique | Accuracy (%) | Precision (%) | Recall (%) | AUC |
|---|---|---|---|---|---|
| Without Balancing | SVM | 83.68 | 0.852 | 0.956 | 0.901 |
| | NB | 76.84 | 0.913 | 0.777 | 0.839 |
| | LR | 83.16 | 0.865 | 0.929 | 0.896 |
| | RF | 82.11 | 0.877 | 0.895 | 0.886 |
| | J48 | 81.84 | 0.870 | 0.902 | 0.886 |
| SMOTE | SVM | 81.25 | 0.834 | 0.882 | 0.857 |
| | NB | 76.72 | 0.892 | 0.723 | 0.799 |
| | LR | 79.56 | 0.827 | 0.858 | 0.842 |
| | RF | 83.41 | 0.871 | 0.868 | 0.870 |
| | J48 | 83.41 | 0.848 | 0.902 | 0.874 |
| Bootstrapping | SVM | 85.79 | 0.856 | 0.983 | 0.915 |
| | NB | 80.79 | 0.937 | 0.807 | 0.868 |
| | LR | 83.42 | 0.861 | 0.939 | 0.898 |
| | RF | 91.05 | 0.946 | 0.939 | 0.942 |
| | J48 | 87.11 | 0.902 | 0.936 | 0.919 |

**Table 7.** Performance of the classifiers using the Relief technique.

| Balancing Technique | Classifier Technique | Accuracy (%) | Precision (%) | Recall (%) | AUC |
|---|---|---|---|---|---|
| Without Balancing | SVM | 84.47 | 0.858 | 0.959 | 0.906 |
| | NB | 75.00 | 0.897 | 0.767 | 0.827 |
| | LR | 83.95 | 0.871 | 0.932 | 0.900 |
| | RF | 82.63 | 0.864 | 0.922 | 0.892 |
| | J48 | 81.58 | 0.867 | 0.902 | 0.884 |
| SMOTE | SVM | 75.22 | 0.855 | 0.736 | 0.791 |
| | NB | 82.33 | 0.850 | 0.878 | 0.864 |
| | LR | 78.23 | 0.842 | 0.811 | 0.826 |
| | RF | 85.13 | 0.854 | 0.926 | 0.888 |
| | J48 | 79.74 | 0.822 | 0.872 | 0.846 |
| Bootstrapping | SVM | 85.53 | 0.855 | 0.980 | 0.913 |
| | NB | 75.26 | 0.901 | 0.767 | 0.828 |
| | LR | 84.74 | 0.872 | 0.943 | 0.906 |
| | RF | 93.16 | 0.944 | 0.970 | 0.957 |
| | J48 | | 0.897 | 0.939 | 0.917 |

Table 5 represents the performance of the classification technique based on different balancing techniques. Without any balancing technique, the LR technique showed the best result, with an accuracy of 83.16%. With SMOTE, the RF technique showed the best result, with an accuracy of 82.97%. With bootstrapping, the RF technique showed the best result, with an accuracy of 90.53%.

Table 6 displays the performance of classifiers utilizing the Gain Ratio technique based on various balancing methods. Without any balancing technique, the J48 technique demonstrated the highest accuracy, at 83.68%. With SMOTE present, the RF and J48

techniques produced the best results, with an accuracy of 83.41%. With bootstrapping, the RF method achieved the best result, with an accuracy of 91.05%.

Table 7 represents the performance of the classification technique based on different balancing techniques. Without any balancing technique, the SVM technique showed the best result, with an accuracy of 84.47%. With SMOTE, the RF technique produced the best results, with an accuracy of 85.13%. With bootstrapping, the RF technique showed the best result, with an accuracy of 93.16%.

## 5. Discussion

Most of the publications included in our review presented high accuracy in classifying individuals with depression based on SMOTE techniques [55,64]. Balancing data by using the SMOTE technique is not the most accurate method for predicting risk of depression. Data balanced using the bootstrapping technique is more accurate than SMOTE.

In this study, we compared SMOTE and bootstrapping techniques. As a result, the accuracy of all the data mining classifiers dramatically increased by combining different feature selection techniques with a balancing technique, as shown in Figure 5. It was discovered that when using the SMOTE technique, the data mining technique's accuracy value was reduced compared to when using the bootstrapping technique. Figure 5c demonstrates the highest degree of accuracy when using Relief selection techniques.

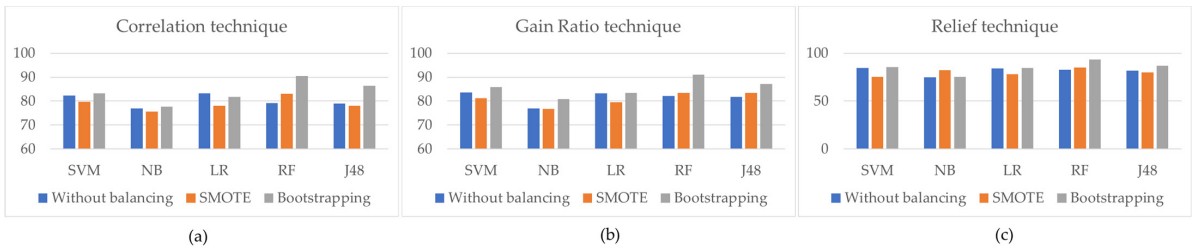

**Figure 5.** Accuracy of the classifiers for different feature selection techniques: (**a**) Comparing outcomes with the correlation method; (**b**) Comparing outcomes with the gain ratio method; and (**c**) Comparing outcomes with the relief method.

The purpose of this study was to identify and create the most significant predictor of depression in students, taking into account socio-demographics, internet addiction, alcohol use disorder, and stress factors, as shown in Table 8.

**Table 8.** Selected attributes involved in each type by using the Relief technique.

|  | Attributes | Description |
|---|---|---|
| Demographic | A5 | Homeland |
|  | A6 | Income adequacy |
|  | A7 | Family status |
| Internet Addiction | B1 | How often do you find that you stay online longer than you intended? |
|  | B14 | How often do you form new relationships with fellow online users? |
|  | B19 | How often do you choose to spend more time online over going out with others? |
| Alcohol Usage | C3 | How often do you have 5 or more drinks on one occasion? |
|  | C9 | Have you or someone else been injured as a result of your drinking? |
|  | C10 | Has a relative, a friend, a doctor, or another health worker been concerned about your drinking or suggested you cut down? |
| Stress | D1 | I have been afraid of making mistakes. |
|  | D15 | I have felt sad and depressed. |
|  | D19 | I have often felt fatigued. |

## 6. Conclusions

Different variables can contribute to the development of depression in a person. This study aimed at identifying the most prevalent risk factors for depression. To assess the risk of depression, a dataset of four types, as well as 380 participants, was compiled. Different feature selection techniques extracted the students' most important socio-demographic, internet addiction, alcohol usage disorder, and stress factors responsible for forming the risk of depression. These feature selection techniques not only increased the training speed of the classifiers, but also improved their ability to accurately screen for depression. This study utilized five different data mining classifiers to determine the presence of depression. By observing the outcomes of the various models presented in this study, it was confirmed that the Random Forest classifier with the Relief feature selection technique was almost the perfect model to predict depression among the participants. It obtained an accuracy of 93.16%.

This work has only predicted the presence of depression in individual students. In the future, this study could be extended to identifying the severity of depression in students. As different socio-demographic and other factors have a big effect on how likely it is that a student will be depressed, this second study could look at the different characteristics of the participants. Several studies show that the performance of a model gets better when different data mining classifiers with balancing techniques and feature selection techniques are used during the data preprocessing steps. In the future, these methods could be used, their results could be compared to those of the current study, and this research could be developed into a more comprehensive depression assessment questionnaire for students than those currently in use.

**Author Contributions:** Conceptualization, W.N. and K.W.; data curation, W.N.; formal analysis, W.N. and K.W.; funding acquisition, W.N.; project administration, W.N. and K.W.; software analysis, W.N. and K.W.; supervision, W.N.; writing—original draft preparation, W.N. and K.W.; writing—review and editing, W.N. and K.W. All authors have read and agreed to the published version of the manuscript.

**Funding:** This research project was financially supported by Mahasarakham University (Grant No: 6603001-2566).

**Informed Consent Statement:** Not applicable.

**Data Availability Statement:** Not applicable.

**Acknowledgments:** This research project was financially supported by Mahasarakham University. The authors are also indebted to Hathaichanok Chompoopong for his English language consulting time and proofreading of the whole paper.

**Conflicts of Interest:** The authors declare no conflict of interest.

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
