# Peer review of "Prediction of Depression for Undergraduate Students Based on Imbalanced Data by Using Data Mining Techniques"

_asi, doi:10.3390/asi5060120_

Round 1
Reviewer 1 Report
The research conducted is inadequately described in this article, and the article is rather sloppily written. The idea itself is good, but the implementation of the research described in this paper has a number of flaws, the major ones of which are listed below.
The procedure used for classification was not described (how it was done - in one or more runs, ...). The only place it is mentioned is in the Abstract, where it is stated that a 10-fold cross-validation was performed. The abstract should be a summary of the article, not the only place in the paper where the authors describe how the research was done!
The authors did not explicitly state whether the questionnaire used was anonymous or not. They should mention this in their paper.
There should be a detailed description of how the research was conducted (e.g., how the authors selected the participants, how the participants completed the questionnaires - on paper, online,... etc.).
It is not clear from the paper how many participants completed the questionnaire, whether there were respondents who did not answer all the questions, etc.
The results section should be more detailed. The authors should also draw some conclusions in this section and not just give out the data from the machine learning application and not mention any conclusions or at least assumptions about those numbers anywhere in the article. For example, it is not explained (or at least assumed) why most of the respondents were female (intentionally, accidentally, because the population is mostly women,...) and if this is an anomaly or expected.
It is not clear how and why informed consent is not applicable to the research described in this paper (as it is stated in this paper) when part of the instrument used for data collection is the "Demographic Information Questionnaire," which is described in one paper as "a personal information questionnaire asking 187 questions about gender, college major, grade level, GPA, home country, adequate income, 188 family status, health status, and history of family depression." Even if this questionnaire is anonymous (the authors did not explicitly state this-although they should if it were), it is quite possible to tell which respondent has a particular questionnaire based on the very detailed personal data collected from the respondents. I believe that with this type of questionnaire, respondents must at least be informed of the details of the research, how their data will be used, whether the data will be collected anonymously or not, and at the very least be asked for their consent to their data being used in this particular research.
In the abstract, the authors write the same word ("Accuracy, accuracy,") twice at the beginning of one sentence.
Subsection 2.6 Imbalance data is in an awkward place in the document. If the authors want to refer to previous research, they should cite some papers that use unbalanced data as opposed to balanced.
The last paragraph of subsection 2.7 has is deeper indentation than the previous paragraphs.
The first sentence in subsection 3.4 is not complete.
The heading of subsection 3.5 is indented incorrectly.
Equation 3 is incorrectly formatted (a closed parenthesis that is not open).
The content of subsection 3.5 describes, from a theoretical point of view, the various algorithms used in classification and does not describe their application in this particular research.
Author Response
Response Letter
Comment |
Response |
Point 1: The procedure used for classification was not described (how it was done - in one or more runs, ...). The only place it is mentioned is in the Abstract, where it is stated that a 10-fold cross-validation was performed. The abstract should be a summary of the article, not the only place in the paper where the authors describe how the research was done! |
We have added additional information to subsection 3.7 |
Point 2: The authors did not explicitly state whether the questionnaire used was anonymous or not. They should mention this in their paper. |
We have added additional information to subsection 3.1. |
Point 3: There should be a detailed description of how the research was conducted (e.g., how the authors selected the participants, how the participants completed the questionnaires - on paper, online,... etc.). |
Described in subsection 3.2 Data Set: number of responses, inclusion and exclusion criteria |
Point 4: It is not clear from the paper how many participants completed the questionnaire, whether there were respondents who did not answer all the questions, etc. |
Described in subsection 3.2 Data Set: number of responses, inclusion and exclusion criteria |
Point 5: The results section should be more detailed. The authors should also draw some conclusions in this section and not just give out the data from the machine learning application and not mention any conclusions or at least assumptions about those numbers anywhere in the article. For example, it is not explained (or at least assumed) why most of the respondents were female (intentionally, accidentally, because the population is mostly women,...) and if this is an anomaly or expected. |
Revised |
Point 6: It is not clear how and why informed consent is not applicable to the research described in this paper (as it is stated in this paper) when part of the instrument used for data collection is the "Demographic Information Questionnaire," which is described in one paper as "a personal information questionnaire asking 187 questions about gender, college major, grade level, GPA, home country, adequate income, 188 family status, health status, and history of family depression." Even if this questionnaire is anonymous (the authors did not explicitly state this-although they should if it were), it is quite possible to tell which respondent has a particular questionnaire based on the very detailed personal data collected from the respondents. I believe that with this type of questionnaire, respondents must at least be informed of the details of the research, how their data will be used, whether the data will be collected anonymously or not, and at the very least be asked for their consent to their data being used in this particular research. |
Revised and described in subsection 3.1 |
Point 7: In the abstract, the authors write the same word ("Accuracy, accuracy,") twice at the beginning of one sentence. |
Revised
|
Point 8: Subsection 2.6 Imbalance data is in an awkward place in the document. If the authors want to refer to previous research, they should cite some papers that use unbalanced data as opposed to balanced. |
Revised and described in subsection 3.2 Line |
Point 9: The last paragraph of subsection 2.7 has is deeper indentation than the previous paragraphs. |
Revised |
Point 10: The first sentence in subsection 3.4 is not complete. |
Revised |
Point 11: The heading of subsection 3.5 is indented incorrectly |
Revised |
Point 12: Equation 3 is incorrectly formatted (a closed parenthesis that is not open). |
Revised |
Point 13: The content of subsection 3.5 describes, from a theoretical point of view, the various algorithms used in classification and does not describe their application in this particular research. |
Revised Summary in Table |
Thank you very much for your kind consideration. We look forward to hearing from you soon.
Warmest regards,
Asst. Prof. Dr Kittipol Wisaeng

Reviewer 2 Report
The submitted manuscript has, in my opinion, several weaknesses.
- The work appears to be not very innovative, as there are many articles in the literature that address the problem of prediction in the case of unbalanced classes and compare dataset balancing techniques.
- Moreover, in the case of diabetes prediction, the most effective technique seems to be ensemble, which could eventually be used after rebalancing the dataset.
- The work should be much improved from the point of view of writing quality.
Author Response
Response Letter
Comment |
Response |
Point 1: The work appears to be not very innovative, as there are many articles in the literature that address the problem of prediction in the case of unbalanced classes and compare dataset balancing techniques. |
We have revised and rewritten based on your feedback. In this study we propose a combined sampling technique to improve the performance of the imbalanced classification of university student depression data. Besides, three different feature selection methods, such as correlation, gain ratio, and relief feature selection algorithms, have been used for extract-ing the most relevant features from the dataset. In our experimental results, we discovered that combining the bootstrapping technique with the relief selection technique under sampling methods enabled the generation of a relatively well-balanced dataset on depression without significant loss of information. Most of the publications included in our review presented high accuracy in classifying individuals with depression participants based on SMOTE techniques. In this study with data balanced using the SMOTE technique is not the most accurate method for predicting risk of depression, whereas data balanced using the bootstrapping tech-nique is more accurate than SMOTE. |
Point 2: Moreover, in the case of diabetes prediction, the most effective technique seems to be ensemble, which could eventually be used after rebalancing the dataset. |
|
Point 3: The work should be much improved from the point of view of writing quality. |
We have revised and rewritten based on your feedback. |
Thank you very much for your kind consideration. We look forward to hearing from you soon.
Warmest regards,
Asst. Prof. Dr Kittipol Wisaeng

Reviewer 3 Report
Dear Authors,
I would like to thank you for the submitted article, which I am able to review. It raises an extremely important issue, which is the prediction of depression occurring among undergraduate youth with the use of innovative data mining techniques. This intersycyplinary approach can certainly be assessed as a significant plus and added value of this study. However, I would like to share with you some of my observations and comments.
1. In my opinion, the abstract in its current form is not legible and does not encourage the reader to read the article. I encourage you to redesign it and reject the exact values ​​in favor of more general (but also factual) sentences that will present the essence of the article, its key conclusions and, above all, its value and novelty. However, I consider the highlighting of the methods used to be valuable and appropriate.
2. On lines 32-37 you do not present the sources you refer to. The reader should know on the basis of which reports and "WHO figures" (by the way, it is too general a term) you present them. You must quote them.
3. Lines 52-59: invite you to introduce the citation source that covers "The Depression Screening Question" along with 9Q / 8Q tests. I also recommend that you present other studies, among which this methodology is considered valuable and important from the point of view of validation of the study. Does source [3] also refer to 9Q / 8Q tests and The Depression Screening Question? More work needs to be done on the fluency of the text here, as in its current form the discussion of these issues may seem like an unnecessary insertion.
4. Rulers 60-63: Please cite specific sources of scientific literature to support the thesis that data mining is widely used in the variety of realms, including medical research and depression-related issues.
5. You have an interestingly divided section 2, where you have made literature review. In view of the structure of this chapter, I encourage you to put forward a research hypothesis after each subsection (2.1., 2.2., Etc.), which will result from an in-depth review of the sources. Currently, the overview of some sections (such as 2.1 and 2.2) is too little thorough.
6. Section 2.1: You merely focus on depression along Thai students and at specific Thai research centers. Comparing the situation in Thailand with other countries would be of great added value.
7. Lines 77-79: Please refer to the source where this definition of depression comes from.
8. Sections 2.6 and 2.7 are not entirely consistent with sections 2.1-2.5. As they stand, sections 2.1-2.5 correspond very neatly to sections 3.1.1-3.1.5 of the methodology section. Chapters 2.6 and 2.7 do not follow this order. Please work on coherence.
9. Question for section 3.2: how was the data retrieved / how was it accessed? Please explain and possibly clarify this issue in the manuscript itself. I am also wondering if at the end of 2022 you do not have more up-to-date data than in 2019?
10. The SMOTE approach and classification algorithms such as SVM and NB were used in an accurate and appropriate manner. Decision trees are presented in an interesting and correct way. Methodologically, the work is in very good condition.
11. It would also be of great value to present whether there are any limitations to this study.
In summary, this work is a very valuable approach that proves that the original adaptation of Bootstrapping with the SMOTE methodology allows for an effective and more effective prediction of depressive behaviors among undergraduates. This is of great value to future researchers. Unfortunately, the current job has some significant shortcomings, especially when it comes to theoretical background. However, after taking into account my comments, I will recommend your article for publication. Thank you in advance for cooperation.
Author Response
Response Letter
Comment |
Response |
Point 1: In my opinion, the abstract in its current form is not legible and does not encourage the reader to read the article. I encourage you to redesign it and reject the exact values in favor of more general (but also factual) sentences that will present the essence of the article, its key conclusions and, above all, its value and novelty. However, I consider the highlighting of the methods used to be valuable and appropriate. |
We have revised and rewritten the abstract based on your feedback. |
Point 2: On lines 32-37 you do not present the sources you refer to. The reader should know on the basis of which reports and "WHO figures" (by the way, it is too general a term) you present them. You must quote them. |
We have revised. |
Point 3: Lines 52-59: invite you to introduce the citation source that covers "The Depression Screening Question" along with 9Q / 8Q tests. I also recommend that you present other studies, among which this methodology is considered valuable and important from the point of view of validation of the study. Does source [3] also refer to 9Q / 8Q tests and The Depression Screening Question? More work needs to be done on the fluency of the text here, as in its current form the discussion of these issues may seem like an unnecessary insertion. |
We have revised the citations to be exact and added a few citations as recommended by the editors. For example, refer to 4, 5, 6, 7, 8, 9, 10, and 11. |
Point 4: Rulers 60-63: Please cite specific sources of scientific literature to support the thesis that data mining is widely used in the variety of realms, including medical research and depression-related issues. |
We have revised the citations to be exact and added a few citations as recommended by the editors. For example, refer to 12 – 14. |
Point 5: You have an interestingly divided section 2, where you have made literature review. In view of the structure of this chapter, I encourage you to put forward a research hypothesis after each subsection (2.1., 2.2., Etc.), which will result from an in-depth review of the sources. Currently, the overview of some sections (such as 2.1 and 2.2) is too little thorough. |
We have revised and described in aim of study in Section 1. |
Point 6: Section 2.1: You merely focus on depression along Thai students and at specific Thai research centers. Comparing the situation in Thailand with other countries would be of great added value. |
|
Point 7: Lines 77-79: Please refer to the source where this definition of depression comes from. |
We have revised.
|
Point 8: Sections 2.6 and 2.7 are not entirely consistent with sections 2.1-2.5. As they stand, sections 2.1-2.5 correspond very neatly to sections 3.1.1-3.1.5 of the methodology section. Chapters 2.6 and 2.7 do not follow this order. Please work on coherence. |
We have revised and described in subsection 3.2 . |
Point 9: Question for section 3.2: how was the data retrieved / how was it accessed? Please explain and possibly clarify this issue in the manuscript itself. I am also wondering if at the end of 2022 you do not have more up-to-date data than in 2019? |
We have revised and described in subsection 3.2: “The survey was conducted in the period between June 2020 and March 2021.” That why we used data retrieved in 2019. |
Point 10: The SMOTE approach and classification algorithms such as SVM and NB were used in an accurate and appropriate manner. Decision trees are presented in an interesting and correct way. Methodologically, the work is in very good condition. |
|
Point 11: It would also be of great value to present whether there are any limitations to this study. |
We have described in section 6 in last paragraph |
Thank you very much for your kind consideration. We look forward to hearing from you soon.
Warmest regards,
Asst. Prof. Dr Kittipol Wisaeng

Round 2
Reviewer 1 Report
I have no further comments.
Reviewer 2 Report
The authors addressed all of the reviewers' concerns.
The revised manuscript can be accepted in its current form.